# Fine-tuning Games:
# Bargaining and Adaptation for General-Purpose Models

*

## ABSTRACT

Major advances in Machine Learning (ML) and Artificial Intelligence (AI) increasingly take the form of developing and releasing general-purpose models. These models are designed to be adapted by other businesses and agencies to perform a particular, domain-specific function. This process has become known as *adaptation* or *fine-tuning*. This paper offers a model of the fine-tuning process where a Generalist brings the technological product (here an ML model) to a certain level of performance, and one or more Domain-specialist(s) adapts it for use in a particular domain. Both entities are profit-seeking and incur costs when they invest in the technology, and they must reach a bargaining agreement on how to share the revenue for the technology to reach the market. For a relatively general class of cost and revenue functions, we characterize the conditions under which the fine-tuning game yields a profit-sharing solution. We observe that any potential domain-specialization will either *contribute*, *free-ride*, or *abstain* in their uptake of the technology, and we provide conditions yielding these different strategies. We show how methods based on bargaining solutions and sub-game perfect equilibria provide insights into the strategic behavior of firms in these types of interactions, and we find that profit-sharing can still arise even when one firm has significantly higher costs than another. We also provide methods for identifying Pareto-optimal bargaining arrangements for a general set of utility functions.

## ACM Reference Format:
. 2018. Fine-tuning Games: Bargaining and Adaptation for General-Purpose Models. In *Proceedings of Make sure to enter the correct conference title from your rights confirmation emai (Conference acronym 'XX)*. ACM, New York, NY, USA, 15 pages. https://doi.org/XXXXXXX.XXXXXXX

## 1 INTRODUCTION

Generative machine-learning (ML) models have garnered a great deal of excitement because they are considered to be *general purpose* [8, 9, 12, 22, 23, 32]. Some have referred to these technologies as *foundation models* [2, 3, 10] because they are designed as massive, centralized models that support potentially many downstream uses. For example, Bommasani et al. [2] write, "a foundation model is itself incomplete but serves as the common basis from which many task-specific models are built via adaptation."

There is palpable excitement about these technologies. But to turn their potential into actual use and impact, one needs to specialize and tweak the technology to particular application domains.

This process takes various names, including *adaptation* [24] and, in some contexts, *fine-tuning* [19, 28, 34].

Notably, the process of fine-tuning a technology involves multiple parties. Technology teams developing ML and Artificial Intelligence (AI) technologies rely on outside entities to adapt, tweak, transfer, and integrate the general-purpose model. This dynamic suggests a latent strategic interaction between producers of a foundational, general-purpose technology and specialists considering whether and how to adopt the technology in a particular context. Understanding this interaction is necessary to study the social and economic consequences of introducing the technology.

This paper brings methods from economic theory to model and analyze the fine-tuning process. We put forward a model of fine-tuning where the interaction between two agents, a generalist and a specialist, determines how they'll bring a general-purpose technology to market (Figure 1). The result of this interaction is a domain-adapted product that offers a certain level of *performance* to consumers, in exchange for a certain level of surplus revenue for the producers. Crucially, the producers must decide how to distribute the surplus, and engage in a bargaining process to do so. An immediate intuition might be to divide this surplus based on contribution to the technology — however, this is one of many potential bargaining solutions, each with different implications for the technology's performance and the distribution of utility. For example, splitting based on contribution can yield a worse-performing technology compared to other bargaining arrangements.

Through this analysis, we discover several general principles that apply not just to today's generative machine learning technologies, but to a potentially wide swath of models that exhibit a similar structure — i.e., developed for general use and adapted to one or more domains. Thus, even as these technologies improve and develop, our proposed model of fine-tuning may continue to describe how they may be adapted for real-world use(s).

Further, as we'll discuss, some of our findings apply to other general-purpose technologies outside machine learning context. For example, cloud computing infrastructure enables a number of consumer-facing services that use web hosting, database services, and other on-demand computing resources. Additive manufacturing (e.g., 3D printing) requires the production of a general-purpose technology that other entities use to create valuable products in particular domains. Digital marketplaces, too, are general market-making technologies that enable specialists (vendors) to sell goods, subject to a bargaining agreement over surplus.

Our main conceptual contribution is modeling the fine-tuning process as a combination of 1) a **multi-stage game** and 2) a **bargaining process** between a general-purpose technology producer and one or more domain specialist(s). Both players bargain over how to share revenue, and each takes a turn contributing to the technology's performance before it reaches the market. Within the set of Pareto-optimal bargaining agreements, we introduce a number of *bargaining solutions* that represent potential arrangements

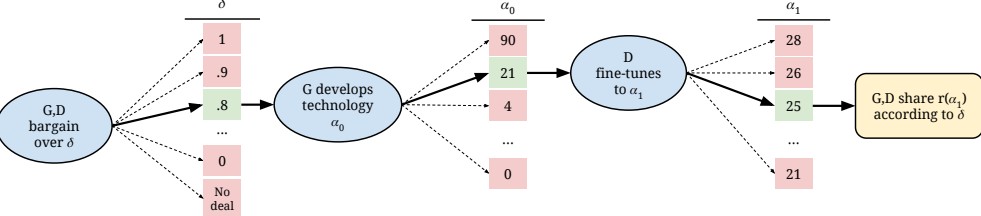

**Figure 1: An illustration of the fine-tuning game. In the first step, players bargain over the revenue-sharing agreement $\delta$. In this example, they agree that G will receive 80% of the revenue and D will receive 20%. In the second step, $G$ develops the technology to performance level $\alpha_0 = 21$. In the third step, $D$ 'fine-tunes' the technology to $\alpha_1 = 25$. If the players collectively receive revenue of 25, they'd share so that $G$ receives 20 and $D$ receives 5.**

for how entities involved in AI's development should distribute profit and effort. These bargaining solutions can be thought of as normative proposals for how to appropriately distribute welfare.

A significant, high-level take-away from our analysis is a characterization of the specialist fine-tuning strategy for any particular domain. We find that any potential adaptor of a technology falls into one of three groups: **Contributors**, who invest effort before selling the technology; **Free-riders**, who sell the technology without investing any additional effort; and **Abstainers**, who do not enter any fine-tuning agreement and opt not to bring the technology to their particular domain. It turns out, using only marginal information about a domain (0th- and 1st-order approximations of cost and revenue), it is possible to reliably determine which strategy the adaptor will take for a notably broad set of scenarios and cost and revenue functions (Section 4.1).

Our analysis consists in deriving the subgame perfect equilibrium strategies, identifying the set of Pareto-optimal bargaining agreements, and then solving for various bargaining solutions proposed by economists. Even in the presence of significant cost differentials, we find bargaining leads to profit-sharing agreements because specialists can leverage their power to exit the deal, reducing the reach of the technology — or, in the case of one specialist, preventing the technology from being produced altogether. For fine-tuning games with a somewhat general set of cost and revenue functions, we develop a method for identifying Pareto-optimal bargains. We find that the Pareto-optimal set is a single interval when there is only one specialist, but may be multiple disjoint intervals in the multi-specialist generalization. The potentially disjoint set arises in cases with multiple specialists because a bargaining deal might be reached with some subset of domain specialists while others abstain.

Some have suggested that scholarship on AI and data-driven technologies focuses predominantly on the technical developments without situating these developments in political economy (though notable exceptions exist) [1, 5, 6, 29, 33]. We propose a model that accounts for the different interests and interactions involved in the development of new, general-purpose AI technology. Our model enables analysis on how these interactions affect market outcomes like performance in practice. Understanding these interactions may also inform future regulation of harms when they arise from generative machine-learning technologies.

## 1.1 Related Work

**Existing approaches to fine-tuning.** Fine-tuning a base model (e.g., a language model [16]) often consists of several steps: (1) gathering and processing domain-specific data, (2) choosing and adjusting the base model's architecture (including number of layers [31] and parameters [26]) and the appropriate objective function [13], (3) Updating the model parameters using techniques like gradient descent or transfer learning, and (4) evaluating the resulting model and refining if necessary. Fine-tuning is an instance of the broader concept of transfer learning [35].

**Existing economic models of general-purpose technology production.** Several lines of work in growth economics address the development and diffusion of general-purpose technologies (or GPTs). See [4] for a survey and [17] for a historic account of electricity and IT as GPTs with major impacts on the US economy. Scholars have examined the effects of factors such as knowledge accumulation, entrepreneurial activity, network effects, and sectoral interactions on the creation of GPTs [15]. The model presented here abstracts away the forces giving rise to the creation of general-purpose technologies, such as LLMs, in the first place, and instead focuses on the later-stage decision of when (or at what performance level) to release the GPT to market for domain-specialization.

Some have suggested that general-purpose technologies create the need for new business models that describe their impact on individual sectors [20]. Gambardella and McGahan [11] proposed a similar model of domain adaptation for general-purpose technology that is based on revenue sharing — however, they do not use bargaining or multi-stage strategy to describe how the technology is developed and brought to market. Our notion of *performance* as it relates to model technologies is inspired by economic models of product innovation [7, 30].

A related—but distinct—body of work is referred to as the hold-up problem [25]. In this literature, two (or more) agents negotiate over an *incomplete* contract and distribute surplus [14]. In these models, after an initial agreement, players are able to re-negotiate and alter parts of the contract, yielding shifts in strategy.

## 2 A MODEL OF FINE-TUNING

In this section, we put forward a model of fine-tuning a data-driven technology for use in a domain-specific context. The technology is developed in two steps: First, a general-purpose producer develops a technology up to a certain level of performance. Then, a domain-specific producer decides whether to adopt the technology, and how

much to invest in the technology to further improve its performance beyond the general-purpose baseline. After these steps, the two entities share a payout.

**Generalist.** Player $G$ (for General-purpose producer) is the first to invest in the technology's performance, and brings the performance level to $\alpha_0 \in \mathbb{R}$. $G$ is motivated to invest in the technology because, ultimately, the technology's performance level determines the revenue $G$ earns.

**Domain Specialist.** After investing in the technology, $G$ can offer the technology to a domain-specialist, denoted $D$, who fine-tunes the model to their specific use case. If $D$ and $G$ enter an agreement, $D$ will invest in improving the technology's performance from $\alpha_0$ to $\alpha_1 \in \mathbb{R}$.

**Revenue and costs.** The technology's *performance*, $\alpha_1$, determines the total revenue that can be gained from fine-tuning the technology in that domain. In particular, we assume there is a monotonic function $r : \mathbb{R}^+ \to \mathbb{R}^+$ such that $r(\alpha_1)$ is the total revenue generated by performance level $\alpha_1$. Unless otherwise specified, we assume $r(\cdot)$ is the identity function, that is, the total revenue brought by technology is $\alpha_1$. The cost associated with producing $\alpha_1$ requires considering the two steps involved with developing the technology: general production and fine-tuning. We say that $G$ faces cost function $\phi_0(\alpha_0) : \mathbb{R}^+ \to \mathbb{R}^+$ to produce a general technology at performance-level $\alpha_0$. $D$ faces cost function $\phi_1(\alpha_1; \alpha_0) : \mathbb{R}^+ \to \mathbb{R}^+$ to bring the technology from performance $\alpha_0$ to performance $\alpha_1$. We assume these cost functions are publicly known. Unless otherwise specified, we also assume $r(0) = 0$, $\phi_0(0) = 0$, and $\phi_1(\alpha_1 = \alpha_0; \alpha_0) = 0$, meaning that not investing in the technology is free and brings in zero revenue.

**The fine-tuning game.** The players are $G$ and $D$. In deciding whether to purchase the technology, $D$ negotiates revenue sharing with $G$. $G$ and $D$ share revenue $r(\alpha_1)$ according to a bargaining parameter $\delta \in [0, 1]$. At the end of the game, $G$ receives $\delta r(\alpha_1)$ in revenue, and $D$ receives $(1 - \delta)r(\alpha_1)$. The model fine-tuning game consists in each player deciding their level of investment and collectively bargaining to decide $\delta$. The game proceeds as follows:

(1) $G$ and $D$ negotiate bargaining coefficient $\delta \in [0, 1]$.
(2) $G$ invests in a general-purpose technology, subject to cost $\phi_0(\alpha_0)$, yielding performance-level $\alpha_0$.
(3) $D$ fine-tunes the technology, subject to cost $\phi_1(\alpha_1; \alpha_0)$, yielding performance-level $\alpha_1$.

The steps of the game are illustrated in Figure 1. Players earn the following utilities, defined as revenue share minus cost:

$$U_G(\delta) := \delta r(\alpha_1) - \phi_0(\alpha_0), \qquad (1)$$

$$U_D(\delta) := (1 - \delta)r(\alpha_1) - \phi_1(\alpha_1; \alpha_0). \qquad (2)$$

If the players do not agree to a feasible bargain $\delta \in [0, 1]$, then the bargaining outcome is referred to as *disagreement*. In this scenario, the generalist receives $d_0$ and the specialist receives $d_1$. We assume, unless otherwise specified, that the disagreement scenario is described by $d_0 = d_1 = 0$.[1]

---

[1]It could be the case that a general-purpose technology producer receives positive payout even if the specialist abstains from a bargain. This case is, essentially, a second 'path' for bringing a general technology to market. We discuss this possibility further in the multi-specialist generalization of our model (Section 4).

## 2.1 Primer on Bargaining Games

Bargaining games are a potentially useful method for computer science research. In this section we include a primer on these methods before demonstrating their use in our model.

A bargain is a process for identifying joint agreements between two or more agents on how to share payoff. The **Bargaining Problem**, formalized by [21], consists of two players that must jointly decide how to share surplus profit. The problem consists of a set of feasible agreements and a 'disagreement' alternative, which specifies the utilities players receive if they do not come to an agreement.

Bargaining solutions are established ways to select among candidate agreements on how to share surplus. Different bargaining solutions, proposed over the years by mathematicians and economists, aim to satisfy certain desiderata like fairness, Pareto optimality, and utility-maximization. Typically, solving for bargaining solutions consists in defining some measure of *joint utility* between players (e.g. take the sum, product, or minimum of the players' utilities). The feasible, Pareto-optimal solution that maximizes this joint utility is known as a **bargaining solution**.

Bargaining solutions are normative: they provide guidelines for how surplus payoffs should be distributed. Solutions are inspired by moral theories like utilitarianism (which aims to maximize the sum of utilities) and egalitarianism (which aims to maximize the worst-off agent). We demonstrate the use of bargaining solutions in the subsequent sections.

## 2.2 Pareto-Optimal Bargains

Our model of the fine-tuning process unfolds in two stages: the first stage is a bargain where the players must jointly agree on $\delta$, and the second stage is a sequential game where the players make decisions individually in order (i.e., $G$ moves first and $D$ moves second). In order to derive solutions, it is important to define *Pareto domination* and *Pareto efficiency*. Once we've defined a few preliminary qualities, we'll state our first finding deriving the set of Pareto-optimal solutions for a general set of cost and revenue functions.

DEFINITION 2.1 (PARETO-DOMINANT AGREEMENTS). *A bargaining agreement $\delta_i$ **Pareto-dominates** an alternative agreement $\delta_j \neq \delta_i$ iff at least one player gains utility by switching from $\delta_j$ to $\delta_i$, and no players lose utility.*

DEFINITION 2.2 (PARETO-OPTIMAL AGREEMENTS). *A **Pareto-optimal** agreement is one where no alternative agreement would improve the utility of one player without decreasing the utility of the other player. In other words, it is an agreement that is not Pareto-dominated by any other agreement.*

DEFINITION 2.3 (STRICTLY UNIMODAL FUNCTION). *A function $f : \mathbb{R} \to \mathbb{R}$ is called a **strictly unimodal function** over a real domain $x \in \mathbf{D}$ if there exists some value $m \in \mathbf{D}$ such that $f$ is strictly increasing $\forall x \leq m$ and $f$ is strictly decreasing $\forall x \geq m$.*

When reasoning about how two agents can jointly reach an agreement, it is useful to start by considering the scenario where one player is *all-powerful*, meaning the bargain is determined solely to maximize one player's utility. The formal definition of this sort of bargaining arrangement is provided below.

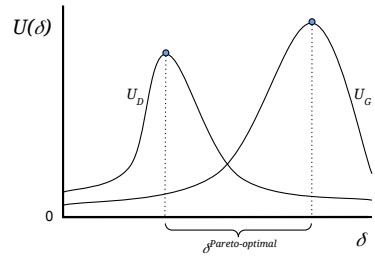

**Figure 2: Example to illustrate Theorem 2.1. For two strictly unimodal, positive utility functions over a bargaining parameter $\delta$, the set of Pareto-optimal bargaining agreements is the interval between their optima.**

DEFINITION 2.4 (POWERFUL-P SOLUTION). *For a given fine-tuning game player $P \in \{G, D\}$, the powerful-P solution is the revenue-sharing agreement $\delta^{Powerful\,P} \in [0, 1]$ that maximizes $P$'s utility:*

$$\delta^{Powerful\,P} = argmax_{\delta \in [0,1]} U_P(\delta).$$

## 2.3 Focus on Unimodal Utilities

We are now in a position to state our first theorem, which characterizes the Pareto-optimal solutions to any fine-tuning game with strictly unimodal utility functions.

THEOREM 2.1. *Consider a fine-tuning game where players bargain over a parameter $\delta$. If the players' utilities are strictly unimodal functions of $\delta$, the set of Pareto-optimal agreements is the interval between their optima $\{\delta^{Powerful\,D}, \delta^{Powerful\,G}\}$, where both players' utilities are greater than the disagreement scenario. If no such interval exists, then disagreement is Pareto-optimal.*

The proof is provided in Appendix 6. To provide some intuition for the proof, consider the range of agreements $\delta$ between the point which maximizes one player's utility (say, $\delta^{Powerful\,D}$) and the point which maximizes the other ($\delta^{Powerful\,G}$). Agreements within this range exhibit a trade-off between the two utilities. Agreements outside this range, however, leave both players worse-off than, e.g., the nearest powerful-P solution, so they are Pareto-dominated. This intuition is illustrated in Figure 2.

Theorem 2.1 applies to a notably broad set of utility functions. To illustrate some of these forms, and for ease of reference, we provide the following immediate corollary:

COROLLARY 2.1. *If $U_D$ is either strictly increasing, strictly decreasing, or strictly concave in $\delta$, and $U_G$ is either strictly increasing, strictly decreasing, or strictly concave in $\delta$, then the set of Pareto-optimal agreements is the interval between their optima $\{\delta^{Powerful\,D}, \delta^{Powerful\,G}\}$, where both players' utilities are greater than the disagreement scenario. If no such interval exists, then disagreement is Pareto-optimal.*

Notice this follows immediately from Theorem 2.1 because any strictly increasing, strictly decreasing or strictly concave function on the interval $\delta \in [0, 1]$ is strictly unimodal on the same interval.

Equipped with the theorem above, solving the fine-tuning game consists of the following steps: (**1**) Use backward induction to solve for $D$ and $G$'s strategies, represented by $\alpha_1^*$ and $\alpha_0^*$, in terms of $\delta$. (**2**) Find the set of Pareto-optimal bargaining agreements $\delta$ between the powerful-D and powerful-G solutions. (**3**) Within the Pareto set,

solve for bargaining agreements that maximize some joint function of the players' utilities.

## 3 ANALYSIS FOR POLYNOMIAL COSTS

Our model applies to general cost and revenue functions, and in Section 4 we provide results at this general level. But to understand how the central parameters of the model interact in closed form, it is also useful to study instantiations of the model with specific functional forms. Accordingly, we show in this section how to solve the model with a set of polynomial cost functions as a paradigmatic instance of convex cost functions, where the marginal costs increase as the technology is improved. Following this, we show how to draw conclusions about the model with general costs.

Thus, in this section, cost functions take the following polynomial function forms:

$$\phi_0(\alpha_0) := c_0 \alpha_0^{k_0}, \tag{3}$$

$$\phi_1(\alpha_1; \alpha_0) := c_1 (\alpha_1 - \alpha_0)^{k_1}. \tag{4}$$

Here, $c_0, c_1 > 0$ since costs should increase with investment, and $k_0, k_1 > 1$, meaning that an incremental improvement grows costlier at higher levels of performance. We will continue to assume that $r(\alpha_1) = \alpha_1$ throughout this section's analysis.

First (3.1), we derive the subgame perfect equilibrium strategies $\alpha_0^*, \alpha_1^*$ for fixed $\delta$. Second (3.2), we find the set of Pareto-optimal revenue-sharing schemes $\delta^{Pareto}$. Reaching a revenue-sharing agreement $\delta^* \in \delta^{Pareto}$ is modeled as a bargaining problem because the players must decide how to share surplus utility. So, third (3.3), we define five potential bargaining solutions: Best-performing-model, Vertical Monopoly, Egalitarian, Nash Bargaining Solution, and Kalai-Smorodinsky. Where possible, we derive closed-form expressions for these solutions. We end by discussing the implications of these different revenue-sharing schemes.

## 3.1 Subgame Perfect Equilibrium for a Given $\delta$

We use backward induction to determine the fine-tuning game's subgame perfect equilibrium (which we will refer to as a 'solution' or 'equilibrium'). Fixing the outcome of the initial negotiation, $\delta$, it is possible establish the following closed-form solution:

THEOREM 3.1. *For a fixed $\delta$, the sub-game perfect equilibrium of the fine-tuning game with polynomial costs yields the following best-response strategies:*

$$\alpha_0^* = \left(\frac{\delta}{k_0 c_0}\right)^{\frac{1}{k_0 - 1}}, \quad \alpha_1^* = \left(\frac{\delta}{k_0 c_0}\right)^{\frac{1}{k_0 - 1}} + \left(\frac{1-\delta}{k_1 c_1}\right)^{\frac{1}{k_1 - 1}}.$$

A proof of the above result is provided in Appendix 7. Notice that the domain-specific performance, $\alpha_1^*$, is equal to the general-purpose performance, $\alpha_0^*$, plus a term, $(\frac{1-\delta}{k_1 c_1})^{\frac{1}{k_1 - 1}}$, independent of the $G$'s choice over $\alpha_0^*$. This is because the cost of marginal improvements for $D$ only depends on the *difference* ($\alpha_1 - \alpha_0$), and is not affected by a large or small initial investment by $G$. Though we assume, in this section, that $D$'s cost is defined solely in terms of marginal improvement, Appendix 7.9 and Section 4 contain findings that generalize beyond this assumption.

As an immediate corollary of Theorem 3.1, we derive players' utilities as a function of $\delta$ alone.

COROLLARY 3.1. *For a fixed bargaining parameter $\delta$, the players' utilities are as follows:*

$$U_G(\delta) = \left(\frac{1}{k_0 c_0}\right)^{\frac{1}{k_0-1}} \left(1 - \frac{1}{k_0}\right) \delta^{\frac{k_0}{k_0-1}} + \left(\frac{1}{k_1 c_1}\right)^{\frac{1}{k_1-1}} \delta(1-\delta)^{\frac{1}{k_1-1}},$$

$$U_D(\delta) = \left(\frac{1}{k_1 c_1}\right)^{\frac{1}{k_1-1}} \left(1 - \frac{1}{k_1}\right)(1-\delta)^{\frac{k_1}{k_1-1}} + \left(\frac{1}{k_0 c_0}\right)^{\frac{1}{k_0-1}} (1-\delta)\delta^{\frac{1}{k_0-1}}.$$

In order to determine the set of Pareto-optimal agreements, we first find that the utility functions derived above are strictly unimodal for all $c_0, c_1$ and $k_0, k_1 \geq 2$.

PROPOSITION 3.1. *In the fine-tuning game with polynomial costs, if $k_0, k_1 \geq 2$, then $U_G$ and $U_D$ are strictly unimodal functions of $\delta \in [0, 1]$.*

The above findings are proven in Appendix 7.3. They suggest that a general set of cost functions yield strictly unimodal utility curves. The set of Pareto-optimal solutions to these games can therefore be identified using Theorem 2.1. It is easy to show that the strict unimodality finding further generalizes to linear combinations of polynomial terms of the form provided in Equations 3 and 4, so long as all exponents are greater than or equal to 2. However, when the condition is not met and $k_0, k_1 < 2$, numerical simulations suggest that there are counter-examples to the strict unimodality property. When the strict unimodality property does not hold, it is still possible to analyze players' strategies—for example, our analysis in Section 4.1 and Appendix 7.9 stands even in cases where utility functions that are not unimodal in $\delta$.

Solving the powerful-$G$, powerful-$D$, vertical monopoly or other bargaining solutions consists in maximizing players' utilities either separately or combined into a joint utility. This is possible once parameters are specified; however, we cannot produce a closed-form expression for the general polynomial case because doing so would require solving for the zeroes of a polynomial of high degree. Therefore, for the remainder of this section, we will demonstrate the solution steps using parameter values $k_0, k_1 = 2$. We call this the case of *quadratic costs*. We choose the quadratic case for clarity and exposition, though we note that other solutions with other parameter values can be calculated using analogous steps.

## 3.2 Pareto-optimal Agreements on $\delta$

We've derived both players' optimal strategies for fixed $\delta$. Now, we consider the process where players agree on a particular value of $\delta$. Since both players must enter an agreement in order for the technology to be viable, the determination of $\delta$ is a two-player bargaining game. We start by solving for the set of Pareto-optimal bargaining agreements, which is the interval between the 'powerful-player' solutions, defined below.

*3.2.1 Powerful-Player Solutions.* As we showed in Theorem 2.1, identifying the 'powerful-player' agreements is important for characterizing the set of Pareto-optimal bargaining solutions. Thus, we begin this section of analysis by solving for the powerful-$G$ and powerful-$D$ solutions (as defined in Definition 2.4).

PROPOSITION 3.2 (POWERFUL-$G$ SOLUTION). *The Powerful-$G$ solution to the model fine-tuning game with quadratic costs is as follows:*

$$\delta^{Powerful\,G} = \begin{cases} \frac{c_0}{2c_0-c_1} & \text{for } c_1 < c_0, \\ 1 & \text{for } c_1 \geq c_0. \end{cases}$$

PROPOSITION 3.3 (POWERFUL-$D$ SOLUTION). *The Powerful-$D$ solution to the model fine-tuning game with quadratic costs is as follows:*

$$\delta^{Powerful\,D} = \begin{cases} 0 & \text{for } c_1 < c_0, \\ \frac{c_1-c_0}{2c_1-c_0} & \text{for } c_1 \geq c_0. \end{cases}$$

Now, using Theorem 2.1 and Proposition 3.1, we can define the set of Pareto-optimal solutions as:

$$\delta^{Pareto} \in \left\{\delta : \delta \leq \delta^{Powerful\,G} \cap \delta \geq \delta^{Powerful\,D}\right\}.$$

A visual representation of these solutions for the fine-tuning game with quadratic costs is given in Figure 4.

## 3.3 Bargaining Solutions to Specify $\delta$

If neither player dominates in a bargain, how do they decide how to share surplus profit? Solutions to bargaining problems tend to find an agreement that maximizes some joint utility function or satisfies certain desirable properties. In this section, we define the various bargaining solutions that the two players could plausibly arrive at within the set of Pareto-optimal solutions. These solutions mostly use a joint utility function to guide the bargaining agreement, as depicted in Figure 3. A visual representation of the bargaining solutions is provided in Figure 4. Definitions and closed-form solutions are provided below, and the proofs and steps yielding the solutions are included in Appendix 7.

**Solution that maximizes the technology's performance.** The first solution we propose presumes the joint goal of the two players is to collectively produce a technology with maximum performance $\alpha_1^*$. There are a few ways to think of this quantity: It is the performance of the technology, and, equivalently, it is also the amount of revenue the two players collect. Though we do not formally specify a social welfare function, the technological performance can be thought of as the total utility offered to society by firms $G$ and $D$.

DEFINITION 3.1 (MAXIMUM-PERFORMANCE SOLUTION). *For the fine-tuning game, the maximum-performance bargaining solution is the feasible revenue-sharing agreement $\delta^{max-\alpha_1^*} \in [0, 1]$ that maximizes the technology's performance $\alpha_1^*$: $\delta^{max-\alpha_1^*} = argmax_{\delta \in [0,1]} \alpha_1^*$.*

PROPOSITION 3.4 (MAXIMUM-$\alpha_1^*$ SOLUTION). *A bargaining solution that maximizes the technology's performance is given by:*

$$\delta^{Max-\alpha_1^*} = \begin{cases} 0 & \text{for } c_1 < c_0, \\ 1 & \text{for } c_1 \geq c_0. \end{cases}$$

**Vertical Monopoly Solution.** A perhaps intuitive approach to bargaining is to choose a revenue-sharing agreement that maximizes the sum of utilities $U_G + U_D$. This solution imagines that the two players are jointly controlled by a single entity who simply wishes to maximize the sum of utility. This solution is known as either the 'vertical monopoly' solution or the 'utilitarian' solution.

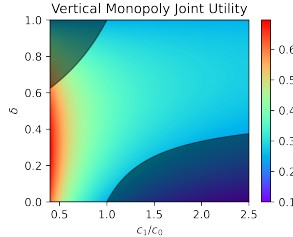 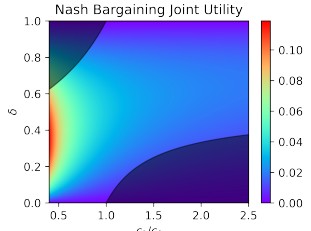 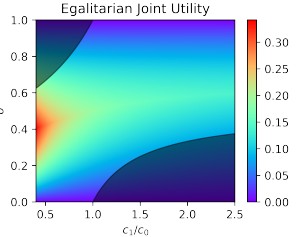

**Figure 3: Various joint-utility functions for finding bargaining solutions. Gray regions are $\delta$ values that are not Pareto-optimal and therefore not candidate bargaining solutions. Color bar scales are defined assuming $c_0 = 1$.**

DEFINITION 3.2 (VERTICAL MONOPOLY SOLUTION). *For the fine-tuning game, the Vertical Monopoly (or 'Utilitarian') Solution is the feasible revenue-sharing agreement $\delta^{VM} \in [0, 1]$ that maximizes the sum of the players' utilities:* $\delta^{VM} = argmax_{\delta \in [0,1]} (U_G(\delta) + U_D(\delta))$.

PROPOSITION 3.5 (VERTICAL MONOPOLY SOLUTION). *The Vertical Monopoly Bargaining Solution to the fine-tuning game with quadratic costs is as follows:*

$$\delta^{Vertical\ Monopoly} = \frac{c_1}{c_1 + c_0}.$$

**Egalitarian Bargaining Solution.** An alternative bargaining approach tries to help the worst-off player. This bargaining solutions is known as the 'egalitarian' solution.

DEFINITION 3.3 (EGALITARIAN BARGAINING SOLUTION). *For the fine-tuning game, the Egalitarian Bargaining Solution is the feasible revenue-sharing agreement $\delta^{Egal.} \in [0, 1]$ that maximizes the minimum of players' utilities:* $\delta^{Egal.} = argmax_{\delta \in [0,1]} (min_{P \in \{G,D\}} (U_P(\delta)))$.

PROPOSITION 3.6 (EGALITARIAN BARGAINING SOLUTION TO THE FINE-TUNING GAME WITH QUADRATIC COSTS). *The Egalitarian Bargaining Solution to the fine-tuning game with quadratic costs is:*

$$\delta^{Egal.} = \frac{-\sqrt{c_0^2 - c_0 c_1 + c_1^2} - c_1 + 2c_0}{3(c_0 - c_1)}.$$

**Nash Bargaining Solution.** The Nash Bargaining solution maximizes the product between the two players' utilities. This arrangement satisfies a number of desiderata, originally laid out by [21].

DEFINITION 3.4 (NASH BARGAINING SOLUTION). *For the fine-tuning game, the Nash Bargaining Solution is the feasible revenue-sharing agreement $\delta^{NBS} \in [0, 1]$ that maximizes the product of the players' utilities:* $\delta^{NBS} = argmax_{\delta \in [0,1]} (U_G(\delta) * U_D(\delta))$.

Though a closed-form solution for quadratic functions is possible, it involves solving the roots of a cubic function and yields a solution that is clunky and uninterpretable. Therefore, we refer the reader to our numerical findings on this solution, which are depicted in Figures 3 and 4.

**Kalai-Smorodinsky Bargaining Solution.** Another solution suggested in economic literature is known as the 'Kalai-Smorodinsky' bargaining solution. This solution equalizes the ratio of maximal gains. More formally:

DEFINITION 3.5 (KALAI-SMORODINSKY BARGAINING SOLUTION). *([18]) For the fine-tuning game, the Kalai-Smorodinsky Bargaining Solution (KSBS) is the feasible revenue-sharing agreement $\delta^{KSBS} \in [0, 1]$ that satisfies the following relation:*

$$\frac{U_G(\delta^{KSBS})}{\max_{\delta \in \delta^{Pareto}} U_G(\delta)} = \frac{U_D(\delta^{KSBS})}{\max_{\delta \in \delta^{Pareto}} U_D(\delta)}.$$

Notice the denominators in the above equation are simply the utilities associated with the powerful-G and powerful-D solutions. Despite this simplifying step, the closed form Kalai-Smorodinsky solution is clunky and uninterpretable, so we omit it from this paper. We refer the reader to our numerical findings on this solution, which are depicted in Figure 4.

## 3.4 Discussion on Bargaining Solutions

Above we solve for a number of bargaining solutions revealing different possible configurations of fine-tuning arrangements. The general technology-producer and the domain specialist each have different optimal arrangements, between which any agreement is Pareto-optimal in the case of polynomial costs.

The first notable take-away is that players do not necessarily opt to maximize their own share of the profit. Even if one player has full control over the bargaining solution, depending on the relative cost of production, they may benefit from a profit-sharing agreement in order to encourage investment by the other player. If bargaining is conceptualized as splitting a pie, one player prefers to cede some portion of the pie if it means the entire pie grows to a size that justifies profit-sharing. This phenomenon arises in real-world settings. For instance, Apple allows third party developers to build software on iPhones. Opening up the tasks of application development to third parties improves consumer experience such that consumers are willing to purchase apps or other capabilities within apps. This additional revenue is then shared between Apple and the developer, leaving Apple with higher profits and a better product. Revenue sharing arises, often, because doing so is lucrative.

Profit-sharing is present even when both players have exceedingly different costs of production (i.e., when $c_i$ approaches 0 or $\infty$). In these limiting instances, we find that the Nash bargaining solution, Kalai-Smorodinski, and Egalitarian solutions all suggest profit-sharing. Only the Utilitarian solution—which models the two players as a vertical monopoly that is centrally controlled—yields the intuitively performance-optimal bargain, where the player with

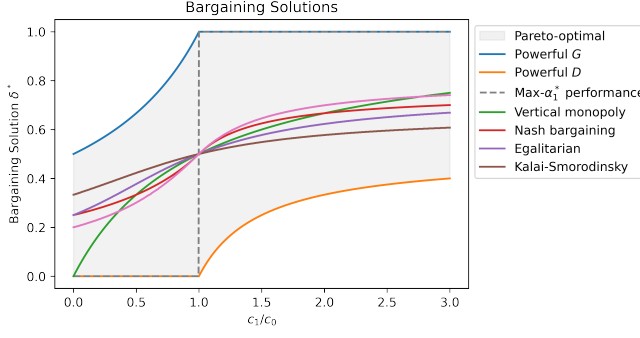

**Figure 4: Bargaining agreements for the fine-tuning game with quadratic costs (i.e., the polynomial cost game where $k_0 = k_1 = 2$). Most bargaining solutions suggest revenue sharing, even where one player faces much higher costs.**

lower costs receives the entire profit. However, the vertical monopoly solution is not always performance-optimal. It underperforms the KSBS when the players face similar costs ($\sim 0.5 < \frac{c_1}{c_0} < 2.5$).

The bargaining solutions are neither binding rules nor descriptive observations; instead, they are *normative*. Identifying joint utility functions can help guide agents towards decisions that serve collective interests. For example, utilitarian and egalitarian solutions offer different visions for the appropriate distribution of welfare. In the same vein, specifying and committing to a *social welfare function* would allow us to identify a bargaining solution that might be referred to as 'socially optimal.' Unsurprisingly, however, specifying social interests in a single function is a difficult undertaking. In any given context, all the relevant interests, requirements, and aims must be accounted for. In our present case, a social welfare function would need to balance the interests of (at least) 1) the technology's producers 2) consumers who value performance and 3) other external stakeholders. The procedure demonstrated in this section (and generalized in the coming sections) provides a road map for a social welfare analysis of the deployment of general-purpose models. Such an analysis might uncover how fine-tuning processes can be configured to serve collective, societal interests.

## 4 MULTIPLE DOMAIN SPECIALISTS

So far, we've modeled the fine-tuning process as a two-player game between a generalist and a specialist. However, an important feature of general-purpose AI models is that they can be developed without fully anticipating the set of possible downstream use-cases. To capture the possibly many use-cases for general-purpose models, in this section, we generalize our model to the case where $n \geq 1$ domain specialists adapt the technology.

**The multi-specialist fine-tuning game.** Consider a game with $n \geq 1$ specialists. The players are $G$, $D_1$, $D_2$,...,$D_n$ and we'll use $i$ to index the specialists. $G$ develops a technology to general performance $\alpha_0$, after which domain specialist $D_i$ specializes the technology to reach performance $\alpha_i$ in their domain. $G$ and $D_i$ share revenue $r_i(\alpha_i)$ according to bargaining parameter $\delta \in [0, 1]$. The bargaining parameter is fixed, which captures common scenarios where the generalist simply has to set a certain pricing agreement

for model access. In other words, $G$ cannot price discriminate depending on domain $i$. The game proceeds as follows:

- Players collectively bargain to decide $\delta \in [0, 1]$.[2]
- $G$ invests in a general-purpose technology yielding performance-level $\alpha_0$ and subject to cost $\phi_0(\alpha_0)$.
- Each specialist $D_i$ may fine-tune the technology by choosing a performance level $\alpha_i$ subject to cost $\phi_i(\alpha_i; \alpha_0)$.

Players' utilities are defined as revenue share minus cost:

$$U_G(\delta) := \sum_i \delta r_i(\alpha_i) - \phi_0(\alpha_0), U_{D_i}(\delta) := (1-\delta)r_i(\alpha_i) - \phi_i(\alpha_i; \alpha_0).$$

If the general-purpose producer does not agree to a feasible bargain $\delta \in [0, 1]$, then all players receive 0 utility. If any particular specialist does not agree to a feasible bargain, this does not preclude other specialists from reaching a deal. Note that some general-purpose technologies might be marketed directly to consumers by the generalist, meaning that reaching a deal with an individual specialist is not, necessarily, needed for $G$ to receive revenue. We believe this scenario can be captured by an additional specialist engaged in a vertical monopoly agreement with $G$.

### 4.1 Domain Specialists' Equilibrium Strategies

When there are potentially many domains where a technology may prove useful or marketable, different strategies around investment levels and fine-tuning can arise. In some domains, a technology may be adopted 'as-is' without significant additional investment or specialization. In other domains, it might be in everyone's interest for a technology to receive significant investment and specialization. Finally, in some domains, a technology might not be viable for any use at all. In this section, we explore the different sorts of cooperation (or non-cooperation) that might arise in domains with different characteristics. Our next general finding is a theorem on the different regimes of domain-specialist strategies, depending on particular attributes of revenue and cost functions.

First, we'll offer a set of relevant definitions to help characterize the different possible regimes of strategies for the specialist. Then, we'll state the formal theorem.

DEFINITION 4.1 (CONTRIBUTOR). *A domain specialist $D_i$ is a **contributor** at the profit-sharing agreement $\delta$ if, given the generalist optimal investment $\alpha_0$ at $\delta$, $D_i$'s optimal strategy is to bring the technology to performance $\alpha_i^* > \alpha_0$.*

DEFINITION 4.2 (FREE-RIDER). *A domain specialist $D_i$ is a **free-rider** at the profit-sharing agreement $\delta$ if, given the generalist optimal investment $\alpha_0$ at $\delta$, $D_i$'s optimal strategy is to enter the deal without improving its performance, so $\alpha_i^* = \alpha_0$.*

DEFINITION 4.3 (ABSTAINER). *A domain specialist $D_i$ is an **abstainer** at the profit-sharing agreement $\delta$ if, given the generalist optimal investment $\alpha_0$ at $\delta$, $D_i$'s optimal strategy is to exit the deal and opt for disagreement.*

Notice that any specialist with any cost and revenue function is inevitably either a contributor, free-rider, or abstainer. These

---

[2]In cases with multiple domain specialists, the bargain over $\delta$ is a multi-player bargaining game. We arrive at bargaining solutions for games with more than two players through a similar process to the two-player case, where a joint utility function is specified and then optimized. For more on collective welfare-driven decisions, see [27].

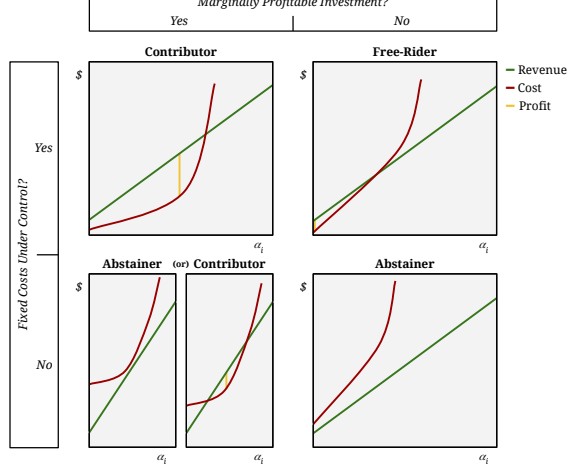

**Figure 5: Examples illustrating Theorem 4.1. When fixed costs are under control and investment is marginally profitable (upper left quadrant), the domain specialist will contribute to the technology. When fixed costs are under control but investment is not marginally profitable (upper right), the domain specialist will free-ride. When fixed costs are too high and investment is marginally costly (lower right), the domain specialist will not bring the technology to market. Finally, when the fixed costs are too high but investment yields marginal returns (lower left), the domain specialist might abstain or contribute, depending on whether revenue sufficiently exceeds cost at any level of investment.**

three regimes span the possible strategies for $D_i$ in the fine-tuning game. Below, we outline conditions that characterize a specialist's strategy depending on their domain's cost and revenue $\{r, \phi_i\}$.

**Theorem 4.1.** *Say a generalist has produced a general-purpose technology operating at performance $\alpha_0$ and available at profit-sharing parameter $\delta$. For any specialist with utility unimodal in $\alpha_i$, the following conditions characterize their strategy, shown in Table 1.*

- *"Fixed Costs Under Control" (FCUC): At zero investment ($\alpha_i = \alpha_0$), the domain specialist $i$'s cost is less than its share of the revenue. Formally, $r_i(\alpha_0) > \frac{1}{1-\delta}\phi_i(\alpha_0)$.*
- *"Marginally Profitable Investment" (MPI): At zero investment ($\alpha_i = \alpha_0$), a marginal investment from the domain specialist $i$ increases its revenue share more than its costs. Formally, $r_i'(\alpha_0) > \frac{1}{1-\delta}\phi_i'(\alpha_0)$.*

| "FCUC" | "MPI" | Type of Specialist |
|:---:|:---:|:---:|
| T | T | Contributor |
| T | F | Free-rider |
| F | T | Contributor or Abstainer* |
| F | F | Abstainer |

**Table 1: Types of specialists. In the third case (*), more information is needed to conclude whether the specialist contributes or abstains.**

A proof of the above theorem is provided in Appendix 8.1. The requirement that specialist utility is unimodal in $\alpha_i$ is, in our view, quite natural and broad. It covers three possible scenarios: 1) specialist utility is increasing with investment, 2) specialist utility is decreasing with investment, or 3) specialist utility increases with investment up to a certain point, beyond which any further investment is not cost-justified.

It is important to note that the three 'regimes' defined in this section can describe a specialist's strategy in either the 1-specialist or multi-specialist fine-tuning game. In the 1-specialist case, the potential strategies describe counter-factuals that depend on the particular cost and revenue functions of the specialist. In the multi-specialist game, all of these regimes are ways of grouping domains and all can exist simultaneously.

One scenario portrayed in Table 1 does not determine cleanly which regime the specialist falls into. In the scenario labeled with an asterisk (*), fixed costs are not under control but it is marginally profitable to invest in the technology. At zero investment, the technology is not ready to bring to market profitably, and it is unclear only from the marginal return on an initial investment whether it is worthwhile for the specialist to invest. In this case, the technology is potentially viable with some non-zero effort spend or, alternatively, not viable for the domain at any level of investment. More information would be needed to conclude whether the specialist would contribute. In particular, if $(1-\delta)r_i(\alpha_i) - \phi_i(\alpha_i)$ has positive real roots (above $\alpha_0$), then we'd conclude the technology is viable.

An illustration of Theorem 4.1 is provided in Figure 5. The indeterminate case contains two possible scenarios, one where the specialist would abstain and one where the specialist would contribute. A neat feature of this result is that these behaviors about particular domain adaptations depend only on attributes about the domain around $\alpha_i = \alpha_0$. It uses only 0th- and 1st-order approximations of $U_{D_i}\big|_{\alpha_i=\alpha_0}$, when the domain has invested no effort.

This theorem perhaps explains why technologies see significant uptake in some domains and not others. It could, potentially, help identify domains that are particularly likely or unlikely to adopt a general purpose technology. It also may explain why some technologies are re-sold without additional investment while others require significant fine-tuning.

## 5 CONCLUSION

Our model provides a starting point for considering the different interests and choices involved in the development of general-purpose models. By putting forward this model, we attempt to invoke the political economy of the development of general AI technologies. These technologies are produced by a number of entities with different interests, and may potentially affect many individuals. This paper models agents' different interests explicitly, and proposes methods for weighing between them in light of societal values.

The work suggests a number of interesting directions for further research. One direction is to identify further general existence results for bargaining solutions with general functions in this model. More broadly, we also believe that formalizing the societal interests involved in AI regulation is an important direction; such a formalism would need to build on an underlying model that contains the economic interests of the firms producing the AI technology. Our model may therefore help form the foundation for such work.

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

# 6 SECTION 2 MATERIALS

## 6.1 Pareto set characterization and Theorem 2.1

PROOF OF THEOREM 2.1. Consider three non-overlapping intervals that collectively span the feasible set $\delta \in [0, 1]$. These intervals are:

(1) $0 \leq \delta < \min(\delta^{\text{Powerful } D}, \delta^{\text{Powerful } G})$
(2) $\min(\delta^{\text{Powerful } D}, \delta^{\text{Powerful } G}) \leq \delta \leq \max(\delta^{\text{Powerful } D}, \delta^{\text{Powerful } G})$
(3) $\max(\delta^{\text{Powerful } D}, \delta^{\text{Powerful } G}) < \delta \leq 1$

We will characterize each of these intervals in turn, finding that intervals (1) and (3) are always Pareto dominated, and interval (2) is characterized by a trade-off in utilities.

(1) Within interval (1), the domain is characterized by $\delta < \min(\delta^{\text{Powerful } D}, \delta^{\text{Powerful } G}) \Rightarrow \delta < \delta^{\text{Powerful } D}$ and $\delta < \delta^{\text{Powerful } G}$. By the definition of a strictly unimodal function (2.3), this means that both utility functions $\{U_D, U_G\}$ are strictly increasing over interval 1. Thus, there exists some quantity $\epsilon > 0$ such that, for any value $\delta$ in interval (1), $U_D(\delta + \epsilon) > U_D(\delta)$ and $U_G(\delta + \epsilon) > U_G(\delta)$. Thus, every potential agreement in interval (1) is Pareto-dominated.

(2) Within interval (2), the domain is characterized by $\min(\delta^{\text{Powerful } D}, \delta^{\text{Powerful } G}) \leq \delta$, and also $\delta \leq \max(\delta^{\text{Powerful } D}, \delta^{\text{Powerful } G})$. If $\delta^{\text{Powerful } D} = \delta^{\text{Powerful } G}$, then the value $\delta = \delta^{\text{Powerful } D} = \delta^{\text{Powerful } G}$ is the unique Pareto-optimal agreement because it is optimal for both players. Otherwise if $\delta^{\text{Powerful } D} \neq \delta^{\text{Powerful } G}$, then interval (2) can be characterized as follows: For one player $P \in \{G, D\}$, the utility $U_P$ one utility function is strictly decreasing because $\delta \geq \delta^{\text{Powerful } P}$ and $U_P(\delta)$ is a strictly unimodal function. For the other player $\{G, D\} \setminus P$, the utility $U_{\{G,D\} \setminus P}$ is strictly increasing because $\delta \leq \delta^{\text{Powerful } \{G,D\} \setminus P}$ and $U_{\{G,D\} \setminus P}(\delta)$ is a strictly unimodal function. Since one player's utility is strictly increasing and the other's is strictly decreasing, any perturbation of $\delta$ within interval (2) constitutes a utility gain for one player and a utility loss for the other. For any value of $\delta$ within this interval, if both players' utilities exceed the disagreement payoff (i.e., positive utility), then $\delta$ is Pareto-optimal.

(3) Within interval (3), the domain is characterized by $\delta > \max(\delta^{\text{Powerful } D}, \delta^{\text{Powerful } G}) \Rightarrow \delta > \delta^{\text{Powerful } D}$ and $\delta > \delta^{\text{Powerful } G}$. By the definition of a strictly unimodal function (2.3), this means that both utility functions $\{U_D, U_G\}$ are strictly decreasing over interval (3). Thus, there exists some quantity $\epsilon > 0$ such that, for any value $\delta$ in interval (3), $U_D(\delta - \epsilon) > U_D(\delta)$ and $U_G(\delta - \epsilon) > U_G(\delta)$. Thus, every potential agreement in interval (3) is Pareto-dominated.

Thus interval (2) is Pareto-efficient among the set of feasible bargaining agreements. □

# 7 SECTION 3 MATERIALS

## 7.1 Subgame perfect equilibrium findings

PROOF OF THEOREM 3.1. We solve the game using backward induction as follows:

First, starting with the last stage (3), we solve for $\alpha_1^*$ given $\alpha_0, \delta, c_1$:

$$\alpha_1^* = \text{argmax}_{\alpha_1} U_D(\alpha_1, \alpha, \delta)$$

$$\Rightarrow \quad \left. \frac{\partial U_D}{\partial \alpha_1} \right|_{\alpha_1 = \alpha_1^*} = 0$$

$$\Rightarrow \quad \left. \frac{\partial}{\partial \alpha_1} \left( (1 - \delta)\alpha_1 - c_1(\alpha_1 - \alpha_0)^{k_1} \right) \right|_{\alpha_1 = \alpha_1^*} = 0$$

$$\Rightarrow \quad (1 - \delta) - k_1 c_1 (\alpha_1^* - \alpha_0)^{k_1 - 1} = 0$$

$$\Rightarrow \quad \alpha_1^* = \alpha_0 + \left( \frac{1 - \delta}{k_1 c_1} \right)^{\frac{1}{k_1 - 1}} . \tag{5}$$

Note that $\frac{\partial^2 U_D}{\partial \alpha_1^2} = -k_1(k_1 - 1)c_1(\alpha_1 - \alpha_0)^{k_1 - 2}$. This quantity is negative as long as $k > 1$, which is assumed. Thus, the $\alpha_1^*$ derived above yields a global maximum of $U_D$.

Second, knowing $D$'s choice of $\alpha_1^*$ above, we solve for $\alpha_0^*$ as follows:

$$\alpha_0^* = \text{argmax}_{\alpha_0} U_G(\alpha_0, \delta)$$

$$\Rightarrow \quad \left. \frac{\partial U_G}{\partial \alpha_0} \right|_{\alpha_0 = \alpha_0^*} = 0$$

$$\Rightarrow \quad \left. \frac{\partial}{\partial \alpha_0} \left( \delta \alpha_1^* - c_0 \alpha_0^{k_0} \right) \right|_{\alpha_0 = \alpha_0^*} = 0$$

$$\Rightarrow \quad \left. \frac{\partial}{\partial \alpha_0} \left( \delta \left( \alpha_0 + \left( \frac{1 - \delta}{k_1 c_1} \right)^{\frac{1}{k_1 - 1}} \right) - c_0 \alpha_0^{k_0} \right) \right|_{\alpha_0 = \alpha_0^*} = 0$$

$$\Rightarrow \quad \left. \frac{\partial}{\partial \alpha_0} \left( \delta \alpha_0 + [\text{const}] - c_0 \alpha_0^{k_0} \right) \right|_{\alpha_0 = \alpha_0^*} = 0$$

$$\Rightarrow \quad \delta - k_0 c_0 (\alpha_0^*)^{k_0 - 1} = 0$$

$$\Rightarrow \quad \alpha_0^* = \left( \frac{\delta}{k_0 c_0} \right)^{\frac{1}{k_0 - 1}} .$$

The second derivative $\frac{\partial^2 U_G}{\partial \alpha^2} = -k_0(k_0 - 1)c_0(\alpha_0)^{k_0 - 2}$. This quantity is negative as long as $k > 1$, which is assumed. Thus, the value of $\alpha_0^*$ derived above yields a global maximum of $U_G$.

Finally, plugging in $\alpha_0^* = \left( \frac{\delta}{k_0 c_0} \right)^{\frac{1}{k_0 - 1}}$ into Equation 5, we obtain the following expression for $\alpha_1^*$ as a function of $\delta$ only:

$$\alpha_1^* = \left( \frac{\delta}{k_0 c_0} \right)^{\frac{1}{k_0 - 1}} + \left( \frac{1 - \delta}{k_1 c_1} \right)^{\frac{1}{k_1 - 1}} .$$

This finishes the proof. □

## 7.2 Utilities as a function of $\delta$

PROOF OF COROLLARY 3.1. Plugging the formulas from Theorem 3.1 into Equation 1, we obtain:

$$U_G = \delta\alpha_1 - \phi_0(\alpha_0)$$

$$= \delta\left(\left(\frac{\delta}{k_0 c_0}\right)^{\frac{1}{k_0-1}} + \left(\frac{1-\delta}{k_1 c_1}\right)^{\frac{1}{k_1-1}}\right) - c_0\left(\left(\frac{\delta}{k_0 c_0}\right)^{\frac{1}{k_0-1}}\right)^{k_0}$$

$$= \delta\left(\frac{\delta}{k_0 c_0}\right)^{\frac{1}{k_0-1}} + \delta\left(\frac{1-\delta}{k_1 c_1}\right)^{\frac{1}{k_1-1}} - c_0\left(\frac{\delta}{k_0 c_0}\right)^{\frac{k_0}{k_0-1}}$$

$$= \left[\left(\frac{1}{k_0 c_0}\right)^{\frac{1}{k_0-1}} - c_0\left(\frac{1}{k_0 c_0}\right)^{\frac{k_0}{k_0-1}}\right]\delta^{\frac{k_0}{k_0-1}}$$

$$\quad + \left(\frac{1}{k_1 c_1}\right)^{\frac{1}{k_1-1}}\delta(1-\delta)^{\frac{1}{k_1-1}}$$

$$= \left(\frac{1}{k_0 c_0}\right)^{\frac{1}{k_0-1}}\left(1-\frac{1}{k_0}\right)\delta^{\frac{k_0}{k_0-1}} + \left(\frac{1}{k_1 c_1}\right)^{\frac{1}{k_1-1}}\delta(1-\delta)^{\frac{1}{k_1-1}}.$$

Plugging the formulas from Theorem 3.1 into Equation 2, we obtain:

$$U_D = (1-\delta)\alpha_1 - \phi_i(\alpha_1;\alpha_0)$$

$$= (1-\delta)\left[\left(\frac{\delta}{k_0 c_0}\right)^{\frac{1}{k_0-1}} + \left(\frac{1-\delta}{k_1 c_1}\right)^{\frac{1}{k_1-1}}\right]$$

$$\quad - c_1\left[\left(\frac{\delta}{k_0 c_0}\right)^{\frac{1}{k_0-1}} + \left(\frac{1-\delta}{k_1 c_1}\right)^{\frac{1}{k_1-1}} - \left(\frac{\delta}{k_0 c_0}\right)^{\frac{1}{k_0-1}}\right]^{k_1}$$

$$= (1-\delta)\left(\frac{\delta}{k_0 c_0}\right)^{\frac{1}{k_0-1}} + (1-\delta)\left(\frac{1-\delta}{k_1 c_1}\right)^{\frac{1}{k_1-1}}$$

$$\quad - c_1\left(\frac{1-\delta}{k_1 c_1}\right)^{\frac{k_1}{k_1-1}}$$

$$= \left(\frac{1}{k_0 c_0}\right)^{\frac{1}{k_0-1}}(1-\delta)\delta^{\frac{1}{k_0-1}} + \left(\frac{1}{k_1 c_1}\right)^{\frac{1}{k_1-1}}(1-\delta)^{\frac{k_1}{k_1-1}}$$

$$\quad - \left(\frac{1}{k_1 c_1}\right)^{\frac{1}{k_1-1}}\left(\frac{1}{k_1}\right)(1-\delta)^{\frac{k_1}{k_1-1}}$$

$$= \left(\frac{1}{k_1 c_1}\right)^{\frac{1}{k_1-1}}\left(1-\frac{1}{k_1}\right)(1-\delta)^{\frac{k_1}{k_1-1}} + \left(\frac{1}{k_0 c_0}\right)^{\frac{1}{k_0-1}}(1-\delta)\delta^{\frac{1}{k_0-1}}.$$

$$\square$$

## 7.3 Utilities are stricly unimodal functions of $\delta$

PROOF OF PROPOSITION 3.1. We'll start by proving $U_G$ is strictly unimodal, and then extend the results to $U_D$.

**Beginning with $U_G$:** The proof relies on the following Lemma:

LEMMA 7.1. *A differentiable continuous function $f(\delta)$ is strictly unimodal over $\delta \in [a,b]$ if the following conditions are met: i) $f'(a) > 0$, ii) $f'$ is concave over the domain.*

PROOF OF LEMMA 7.1. By the definition of strict unimodality, we can conclude a function $f(\delta)$ is strictly unimodal over an interval $[a,b]$ if one of the following properties hold: 1) $f'(\delta) > 0 \; \forall \; \delta \in$

[0, 1], meaning the function is strictly increasing over the interval, or 2) For some value $c \in (a,b)$, the function is strictly increasing for values $[a,c)$ and strictly decreasing for values $(c,b]$. Notice that, so long as condition (**i**) holds (i.e., the function starts out strictly increasing at $a$), the function is strictly unimodal as long as its derivative crosses the $f' = 0$ axis at *no more than one* point in $[a,b]$. So, the remainder of the proof finds a contradiction when we assume conditions (**i**) and (**ii**) and that there are two values in $(a,b)$ for which $f' = 0$.

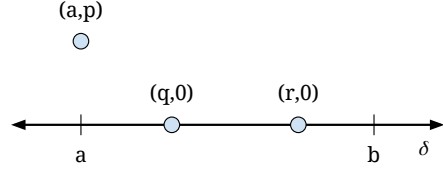

**Figure 6: Illustration of the proof for Lemma 7.1. If the derivative of a function $f$ is positive at $a$ and concave, it cannot cross the axis more than 1 time, meaning $f$ is strictly unimodal.**

Consider the curve $f' = \frac{df}{d\delta}$. We specify a point on this curve using $\{(x,y)|x=\delta, y=\frac{df}{d\delta}\}$. Given condition (**i**), There is some point $x_0 = (a,p)$ where $p > 0$ and $x_0 \in \frac{df}{d\delta}$. Assume for the sake of contradiction that there are two points $x_1 = (q,0), x_2 = (r,0)$ where $x_1, x_2 \in \frac{df}{d\delta}$ and $a < q < r < b$ (without loss of generality). We can plug these three points into the definition of concavity and find our contradiction: First, notice $l = \frac{q-a}{r-a} \in (0,1)$ because $a < q < r$. Next, plugging in the definition of concavity:

$$\frac{df}{d\delta}((1-l)a + lr) \geq (1-l)\frac{df}{d\delta}(a) + l\frac{df}{d\delta}(r)$$

$$\frac{df}{d\delta}\left(\left(1-\frac{q-a}{r-a}\right)a + \frac{q-a}{r-a}r\right) \geq \left(1-\frac{q-a}{r-a}\right)[p] + \frac{q-a}{r-a}[0]$$

$$\frac{df}{d\delta}\left(a - \frac{q-a}{r-a}a + \frac{q-a}{r-a}r\right) \geq p - \frac{q-a}{r-a}p$$

$$\frac{df}{d\delta}\left(a + (r-a)\frac{q-a}{r-a}\right) \geq p - \frac{q-a}{r-a}p$$

$$\frac{df}{d\delta}\left(a + (r-a)\frac{q-a}{r-a}\right) \geq p - \frac{q-a}{r-a}p$$

$$\frac{df}{d\delta}(a + (q-a)) \geq p - \frac{q-a}{r-a}p$$

$$\frac{df}{d\delta}(q) \geq p - \frac{q-a}{r-a}p$$

$$0 \geq p - \frac{q-a}{r-a}p > 0.$$

Hence the contradiction: We know $p - \frac{q-a}{r-a}p > 0$ is strictly greater than 0 because $p > 0$ and $l \in (0,1)$. Thus a function characterized by conditions (**i**) and (**ii**) cannot contain these three points. This concludes Lemma 7.1's proof: If $f'$ is concave, and starts positive at $a$, it must cross the axis at most once meaning $f$ is unimodal.

□

Now, we prove the utilities are unimodal by showing that $U_G(\delta)$ satisfies both conditions in Lemma 7.1 for the domain $\delta \in [0, 1]$.

We first differentiate $U_G$ with respect to $\delta$.

$$\frac{\partial U_G}{\partial \delta} = \frac{\partial}{\partial \delta}\left[\left(\frac{1}{k_0 c_0}\right)^{\frac{1}{k_0-1}}\left(1 - \frac{1}{k_0}\right)\delta^{\frac{k_0}{k_0-1}} + \left(\frac{1}{k_1 c_1}\right)^{\frac{1}{k_1-1}}\delta(1-\delta)^{\frac{1}{k_1-1}}\right]$$

$$= \frac{\partial}{\partial \delta}\left[A\delta^{\frac{k_0}{k_0-1}} + B\delta(1-\delta)^{\frac{1}{k_1-1}}\right],$$

where $A := \left(\frac{1}{k_0 c_0}\right)^{\frac{1}{k_0-1}}\left(1 - \frac{1}{k_0}\right) > 0$ and $B := \left(\frac{1}{k_1 c_1}\right)^{\frac{1}{k_1-1}} > 0$. $A$ is positive as long as $k_0 > 1$ and $c_0 > 0$, which is given. $B$ is positive as long as $k_1 > 1$ and $c_1 > 0$, which is given. Continuing:

$$= A\left(\frac{k_0}{k_0-1}\right)\delta^{\frac{1}{k_0-1}}$$

$$+ B\left[-\delta\left(\frac{1}{k_1-1}\right)(1-\delta)^{\frac{1}{k_1-1}-1} + (1-\delta)^{\frac{1}{k_1-1}}\right]$$

$$= A\left(\frac{k_0}{k_0-1}\right)\delta^{\frac{1}{k_0-1}} + \frac{(1-\delta)^{\frac{1}{k_1-1}-1}}{k_1-1}[-\delta + (k_1-1)(1-\delta)]$$

$$= A\left(\frac{k_0}{k_0-1}\right)\delta^{\frac{1}{k_0-1}} + \frac{(1-\delta)^{\frac{1}{k_1-1}-1}}{k_1-1}[k_1 - k_1\delta - 1 + \delta - \delta]$$

$$= A\left(\frac{k_0}{k_0-1}\right)\delta^{\frac{1}{k_0-1}} + \frac{(1-\delta)^{\frac{1}{k_1-1}-1}}{k_1-1}[k_1(1-\delta) - 1]$$

$$\frac{\partial U_G}{\partial \delta} = A\left(\frac{k_0}{k_0-1}\right)\delta^{\frac{1}{k_0-1}} + B\left(\frac{k_1}{k_1-1}\right)(1-\delta)^{\frac{1}{k_1-1}}$$

$$- B\left(\frac{1}{k_1-1}\right)(1-\delta)^{\frac{1}{k_1-1}-1}.$$

Now we can show the first condition (a) in Lemma 7.1 holds:

$$\left.\frac{\partial U_G}{\partial \delta}\right|_{\delta=0} = [0] + B\left(\frac{k_1}{k_1-1}\right) - B\left(\frac{1}{k_1-1}\right) = B\left(\frac{k_1-1}{k_1-1}\right) = B > 0.$$

To show the second condition (b) in Lemma 7.1 holds, we perform the second-derivative test, which requires differentiating the function two more times:

$$\frac{\partial^3 U_G}{\partial \delta^3} = A\left(\frac{k_0}{k_0-1}\right)\left(\frac{1}{k_0-1}\right)\left(\frac{1}{k_0-1}-1\right)\delta^{\frac{1}{k_0-1}-2}$$

$$+ B\left(\frac{k_1}{k_1-1}\right)\left(\frac{1}{k_1-1}\right)\left(\frac{1}{k_1-1}-1\right)(1-\delta)^{\frac{1}{k_1-1}-2}$$

$$- B\left(\frac{1}{k_1-1}\right)\left(\frac{1}{k_1-1}-1\right)\left(\frac{1}{k_1-1}-2\right)(1-\delta)^{\frac{1}{k_1-1}-3}.$$

The above expression is never positive. First, notice all three coefficients are less than or equal to zero:

- $A(\frac{k_0}{k_0-1})(\frac{1}{k_0-1})(\frac{1}{k_0-1}-1)$ is the product of one negative and otherwise non-negative numbers: Given $k_0 \geq 2$, observe $A > 0$, $(\frac{k_0}{k_0-1}) > 0$, $(\frac{1}{k_0-1}) > 0$, $(\frac{1}{k_0-1}-1) \leq 0$.
- $B(\frac{k_1}{k_1-1})(\frac{1}{k_1-1})(\frac{1}{k_1-1}-1)$ is the product of one negative and otherwise non-negative numbers: Given $k_1 \geq 2$, observe $B > 0$, $(\frac{k_1}{k_1-1}) > 0$, $(\frac{1}{k_1-1}) > 0$, $(\frac{1}{k_1-1}-1) \leq 0$.

- $-B(\frac{1}{k_1-1})(\frac{1}{k_1-1}-1)(\frac{1}{k_1-1}-2)$ is the product of three negative and otherwise non-negative numbers: Given $k_1 \geq 2$, observe $-B < 0$, $(\frac{1}{k_1-1}) > 0$, $(\frac{1}{k_1-1}-1) \leq 0$, $(\frac{1}{k_1-1}-1) < 0$.

Second, notice all three expressions of $\delta$ are defined and positive on the interval $(0, 1)$:

- $\delta^{\frac{1}{k_0-1}-2}$ is positive and defined $\forall \delta > 0$.
- $(1-\delta)^{\frac{1}{k_1-1}-2}$ is positive and defined $\forall \delta < 1$.
- $(1-\delta)^{\frac{1}{k_1-1}-3}$ is positive and defined $\forall \delta < 1$.

Every term in our derived expression for $\frac{\partial^3 U_G}{\partial U_G^3}$ is non-positive. Thus the function $\frac{\partial U_G}{\partial U_G}$ is concave satisfying condition (b) in Lemma 7.1. This completes the proof that $U_G$ is unimodal.

**Moving on to $U_D$:** Notice the formulation of $U_G$ in Corollary 3.1 is almost exactly the same functional form as $U_D$. If we define a variable $\gamma = (1-\delta)$, we can use the identical proof completed above to show $U_D$ is unimodal in $\gamma$. Since we prove unimodality on the interval $[0, 1]$, a function defined over $\gamma \in [0, 1]$ is simply a function of $\delta \in [0, 1]$ reflected over the vertical line $\delta = 0.5$. A transform that reflects a univariate function over the vertical line passing through the midpoint of its domain preserves strict unimodality. □

## 7.4 Powerful-G Bargaining Solution

PROOF OF PROPOSITION 3.2. The powerful-$G$ solution is the solution $\delta^{Powerful\,G}$ that maximizes $U_G$ over the feasible set of $\delta \in [0, 1]$:

$$\delta^{Powerful\,G} = \text{argmax}_\delta U_G(\delta)$$

$$\Rightarrow \quad \frac{\partial U_G}{\partial \delta} = 0$$

$$\Rightarrow \quad \frac{\partial}{\partial \delta}\left[\frac{\delta^2}{4c_0} + \frac{\delta}{2c_1} - \frac{\delta^2}{2c_1}\right] = 0 \quad Corr.\ 3.1$$

$$\Rightarrow \quad \frac{\delta}{2c_0} + \frac{1}{2c_1} - \frac{\delta}{c_1} = 0$$

$$\Rightarrow \quad \delta\left(\frac{1}{2c_0} - \frac{1}{c_1}\right) = -\frac{1}{2c_1}$$

$$\Rightarrow \quad \delta = -\frac{1}{2c_1}\left(\frac{c_1-2c_0}{2c_1 c_0}\right)^{-1}$$

$$\Rightarrow \quad \delta = \frac{c_0}{2c_0-c_1}.$$

The second partial derivative $\frac{\partial^2 U_G}{\partial \delta^2} = \frac{1}{2c_0} - \frac{1}{c_1}$, which is negative as long as $0 < \frac{c_1}{c_0} < 2$. Since there is only one root, the derived equation is a global maximum for $0 < c_1 < 2c_0$. However, notice that the derived expression is only feasible for the values $c_1 \leq c_0$, since the value $\delta$ must be in the range $[0, 1]$ (Specialist would not take a negative share of the profit). Thus, $\delta^{Powerful\,G} = \frac{1}{2-c_1}$ for $0 < c_1 < c_0$.

The remainder of the proof will show that, for $c_1 \geq c_0$, within the feasible set $0 \leq \delta \leq 1$, $\delta = 1$ maximizes $U_G$. We'll do so by showing that the partial derivative $\frac{\partial U_G}{\partial \delta}$ is non-negative for all $c_1 \geq c_0$ and $0 \leq \delta \leq 1$. Assume for sake of contradiction:

$$\frac{\partial U_G}{\partial \delta} < 0$$

$$\Rightarrow \quad \frac{\delta}{2c_0} + \frac{1}{2c_1} - \frac{\delta}{c_1} < 0$$

$$\Rightarrow \quad \delta\left(\frac{1}{2c_0} - \frac{1}{c_1}\right) + \frac{1}{2c_1} < 0$$

$$\Rightarrow \quad \delta\left(\frac{c_1 - 2c_0}{2c_1 c_0}\right) + \frac{c_0}{2c_1 c_0} < 0$$

$$\Rightarrow \quad \delta(c_1 - 2c_0) + c_0 < 0 \quad \text{because } c_1, c_0 > 0.$$

$$\Rightarrow \quad \delta(c_1 - 2c_0) < -c_0.$$

Notice the resulting expression is only met when $\delta < 0$ or $c_1 \leq c_0$. However, we're given $\delta \in [0, 1] \cap c_1 \geq c_0$. $\qquad \square$

### 7.5 Powerful-D Bargaining Solution

PROOF OF PROPOSITION 3.3. The powerful-$D$ solution is the solution $\delta^{Powerful\, D}$ that maximizes $U_D$ over the feasible set:

$$\delta^{Powerful\, D} = \text{argmax}_\delta U_D(\delta)$$

$$\Rightarrow \quad \frac{\partial U_D}{\partial \delta} = 0$$

$$\Rightarrow \quad \frac{\partial}{\partial \delta}\left[\left(\frac{1}{4c_1}\right) + \left(\frac{1}{2c_0} - \frac{1}{2c_1}\right)\delta + \left(\frac{1}{4c_1} - \frac{1}{2c_0}\right)\delta^2\right] = 0 \quad \text{Corr. 3.1}$$

$$\Rightarrow \quad \left(\frac{1}{2c_0} - \frac{1}{2c_1}\right) + 2\left(\frac{1}{4c_1} - \frac{1}{2c_0}\right)\delta = 0$$

$$\Rightarrow \quad \left(\frac{1}{2c_1} - \frac{1}{c_0}\right)\delta = \frac{1}{2c_1} - \frac{1}{2c_0}$$

$$\Rightarrow \quad \delta = \frac{\left(\frac{c_0 - c_1}{2c_0 c_1}\right)}{\left(\frac{c_0 - 2c_1}{2c_0 c_1}\right)}$$

$$\Rightarrow \quad \delta = \frac{c_1 - c_0}{2c_1 - c_0}.$$

The second partial derivative $\frac{\partial^2 U_D}{\partial \delta^2} = \frac{1}{2c_1} - \frac{1}{c_0}$, which is negative as long as $2c_1 > c_0$. Since there is only one root, the derived equation is a global maximum for $2c_1 > c_0$. However, notice that the derived expression is only feasible for the values $c_1 \geq c_0$, since the value $\delta$ must be in the range $[0, 1]$ (Generalist would not take a negative share of the profit). Thus, $\delta^{Powerful\, D} = \frac{c_1 - c_0}{2c_1 - c_0}$ for $c_1 \geq c_0$.

The remainder of the proof will show that, for $c_1 < c_0$, within the feasible set $0 \leq \delta \leq 1$, $\delta = 0$ maximizes $U_D$. We'll do so by showing that the partial derivative $\frac{\partial U_D}{\partial \delta} \leq 0$ for all $0 < c_1 < 1$ and $0 \leq \delta \leq 1$. Assume for sake of contradiction:

$$\frac{\partial U_D}{\partial \delta} > 0$$

$$\Rightarrow \quad \left(\frac{1}{2c_0} - \frac{1}{2c_1}\right) + \left(\frac{1}{2c_1} - \frac{1}{c_0}\right)\delta > 0$$

$$\Rightarrow \quad \left(\frac{1}{2c_1} - \frac{1}{c_0}\right)\delta > \frac{1}{2c_1} - \frac{1}{2c_0}$$

$$\Rightarrow \quad \left(\frac{c_0 - 2c_1}{2c_0 c_1}\right)\delta > \frac{c_0 - c_1}{2c_0 c_1}$$

$$\Rightarrow \quad (c_0 - 2c_1)\delta > c_0 - c_1 \quad \text{because } c_0, c_1 > 0.$$

We show the contradiction $\forall \frac{c_1}{c_0} \in (0, 1)$ (equivalently, every scenario where $0 < c_1 \leq c_0$):

- For $\frac{1}{2} < \frac{c_1}{c_0} < 1$, the final step implies $\delta > \frac{c_1 - c_0}{2c_1 - c_0}$, which contradicts the global optimum finding above.
- For $0 < \frac{c_1}{c_0} < \frac{1}{2}$, the final step implies $\delta \leq \frac{c_1 - c_0}{2c_1 - c_0}$. But notice the right-hand-side must be negative, contradicting the given range $\delta \in [0, 1]$.
- Finally, for $\frac{c_1}{c_0} = \frac{1}{2}$, the final step implies $0 > \frac{1}{2}$.

Thus we've established the contradiction. $\qquad \square$

### 7.6 Maximum-performance bargaining solution

PROOF OF PROPOSITION 3.4. We will show that within the feasible region $\delta \in [0, 1], c_1 > 0$, the following three properties hold:

(1) $\frac{\partial \alpha_1}{\partial \delta}(c_1) < 0 \ \forall \ c_1 < c_0$

(2) $\frac{\partial \alpha_1}{\partial \delta}(c_1) > 0 \ \forall \ c_1 > c_0$

(3) $\frac{\partial \alpha_1}{\partial \delta}(c_1) = 0 \ \text{for} \ c_1 = c_0$

First, we differentiate our expression for $\alpha_1(\delta; c_0, c_1)$ with respect to $\delta$, using the expression attained in Theorem 3.1:

$$\frac{\partial \alpha_1}{\partial \delta}$$

$$= \quad \frac{\partial}{\partial \delta}\left[\frac{\delta}{2c_0} + \frac{1-\delta}{2c_1}\right]$$

$$= \quad \frac{1}{2c_0} - \frac{1}{2c_1}.$$

Now notice each of the three properties are satisfied in turn:

(1) For $c_1 < c_0$, $\frac{\partial \alpha_1}{\partial \delta} = \frac{1}{2c_0} - \frac{1}{2c_1} < 0$.

(2) For $c_1 > c_0$, $\frac{\partial \alpha_1}{\partial \delta} = \frac{1}{2c_0} - \frac{1}{2c_1} > 0$.

(3) For $c_1 = c_0$, $\frac{\partial \alpha_1}{\partial \delta} = \frac{1}{2c_0} - \frac{1}{2c_1} = 0$. $\qquad \square$

### 7.7 Vertical monopoly bargaining solution

PROOF OF PROPOSITION 3.5. The vertical monopoly or "utilitarian" solution is the solution that maximizes the sum of utilities:

$$\delta^{Vertical\, Monopoly} = \text{argmax}_\delta U_G(\delta) + U_D(\delta)$$

$$\Rightarrow \quad \frac{\partial}{\partial \delta}\left(U_G(\delta) + U_D(\delta)\right) = 0$$

$$\Rightarrow \quad \frac{\delta}{2c_0} + \frac{1}{2c_1} - \frac{\delta}{c_1} + \left(\frac{1}{2c_0} - \frac{1}{2c_1}\right) + 2\left(\frac{1}{4c_1} - \frac{1}{2c_0}\right)\delta = 0 \quad \text{Corr. 3.1}$$

$$\Rightarrow \quad \delta\left(\frac{1}{2c_0} - \frac{1}{c_1} + \frac{1}{2c_1} - \frac{1}{c_0}\right) = -\frac{1}{2c_1} - \frac{1}{2c_0} + \frac{1}{2c_1}$$

$$\Rightarrow \quad \delta(c_1 - 2c_0 + c_0 - 2c_1) = -c_1$$

$$\Rightarrow \quad \delta = \frac{c_1}{c_1 + c_0}.$$

The second partial derivative is $\frac{\partial^2 U_G}{\partial \delta^2} = -\frac{1}{2c_1} - \frac{1}{2c_0}$ which is negative for any $c_0, c_1 > 0$, meaning $\delta^{Vertical\, Monopoly} = \frac{c_1}{c_1 + c_0}$ maximizes the sum of utilities. $\qquad \square$

### 7.8 Egalitarian bargaining solution

PROOF OF PROPOSITION 3.6. The Kalai (egalitarian) solution $\delta^{Egal.}$ is the solution that maximizes the minimum utility among players.

First, observe that if there exists a point in the Pareto solution set where the two utilities are equal, this point must be the egalitarian solution. Pareto means that an increase in any player's utility must correspond to a decrease in another player's utility. If a solution within the Pareto set equalizes utilities, then any other solution

must inevitably trade off one player's utility for the other's, meaning any alternative solution would yield a lower utility for at least one player.

So, setting the utilities equal we get:

$$U_G(\delta) = U_D(\delta)$$

$$\frac{\delta^2}{4c_0} + \frac{\delta}{2c_1} - \frac{\delta^2}{2c_1} = \frac{1}{4c_1} + \frac{\delta}{2c_0} - \frac{\delta}{2c_1} + \frac{\delta^2}{4c_1} - \frac{\delta^2}{2c_0} \quad \text{Corr. 3.1}$$

$$\delta^2(3c_1 - 3c_0) + \delta(4c_0 - 2c_1) - c_0 = 0$$

Plugging into the quadratic formula, we get two candidate solutions:

$$\delta^{Egal.} \stackrel{?}{=} \left\{ \frac{\sqrt{c_0^2 - c_0 c_1 + c_1^2} - c_1 + 2c_0}{3(c_0 - c_1)}, \frac{-\sqrt{c_0^2 - c_0 c_1 + c_1^2} - c_1 + 2c_0}{3(c_0 - c_1)} \right\}$$

Notice that the first of these solutions, for $c_0, c_1 > 0$, is not in the feasible set $0 \leq \delta \leq 1$. Thus, the Egalitarian solution is given by:

$$\delta^{Egal.} = \frac{-\sqrt{c_0^2 - c_0 c_1 + c_1^2} - c_1 + 2c_0}{3(c_0 - c_1)}.$$

□

## 7.9 One-player findings for general revenue and costs

The fine-tuning game may be defined for any cost and revenue functions $\phi_0, \phi_1, r$. We demonstrate in Section 3 that closed-form solutions are attainable for certain polynomial function forms. In this section, we provide *general* results that suggest the existence of solutions for a broad set of cost and revenue functions. We put forward an existence finding that suggests meaningful cooperation (profit sharing agreement without free-riding) is viable in a swath of fine-tuning games. Our result is that for any non-decreasing cost and revenue functions, as long as $r'(0) > \lambda_0 \phi_0'(0)$ and $r'(\alpha_0) > \lambda_1 \phi_1'(\alpha_0)$ where $\lambda_0, \lambda_1 \geq 2$, there exists a profit-sharing solution that (a) Pareto-dominates disagreement and (b) does not lead to free-riding.

Before we state the theorem formally, we have to define free riding for the fine-tuning game:

**Definition 7.1 (Free riding).** *A solution to the fine-tuning game $\delta_i \in [0, 1]$ exhibits **free riding** if at least one player receives profit without investing any effort in improving the technology. That is, either $\alpha_0 = 0$ or $\alpha_1 = \alpha_0$.*

The formal theorem statement is below:

**Theorem 7.1 (Revenue sharing solution for the fine-tuning game).** *Consider any fine-tuning game where $r(\alpha_1)$, $\phi_0(\alpha_0)$, and $\phi_1(\alpha_1)$ are non-decreasing. If the following two marginal conditions are met:*

- *Condition 1: $r'(0) > \lambda_0 \phi_0'(0)$ where $\lambda_0 \geq 2$,*
- *Condition 2: $r'(\alpha_0) > \lambda_1 \phi_1'(\alpha_0)$ where $\lambda_1 \geq 2$,*

*Then there exists a solution $\delta^*$ to the fine-tuning game with the following properties: (A) Players share revenue $0 < \delta^* < 1$. (B) Players do not free ride. (C) $\delta^*$ Pareto-dominates disagreement.*

**Proof of Theorem 7.1.** We prove this theorem via a sequence of Lemmas.

**Lemma 7.2.** *If $r'(0) > \lambda_0 \phi_0'(0)$ for a constant $\lambda_0 \geq 2$, then there exists a set $A^* \subseteq (0, 1)$ such that $\frac{1}{2} \in A^*$ and for all $\delta \in A^*$, $\alpha_0^*(\delta) > 0$, $U_G(\delta) > 0$.*

Let's presume $\delta^* = \frac{1}{2}$. If we can show that this solution yields $\alpha_0^* > 0$, then this means the generalist's utility is greater than 0 for investing some non-zero effort spend. $\alpha_0^*$ is the value of $\alpha_0$ that maximizes $U_G$, so if $U_G$ has positive slope at $\alpha_0 = 0$, then $\alpha_0^* > 0$.

Notice that this positive-utility outcome is met as long as $\frac{\partial U_G}{\partial \alpha_0}\big|_{\alpha_0=0} > 0$. This condition would necessarily mean that there exists some positive $\alpha_0 > 0$ which maximizes $U_G$. So, formally:

$$r'(0) > 2\phi_0'(0)$$

$$\frac{1}{2}r'(0) - \phi_0'(0) > 0$$

$$\frac{\partial}{\partial \alpha_0}(\delta r - \phi_0)\Big|_{\alpha_0=0} > 0$$

$$\frac{\partial U_G}{\partial \alpha_0}\Big|_{\alpha_0=0} > 0.$$

Notice that this inequality is met as long as $r'(0) > \lambda_0 \phi_0'(0)$ where $\lambda_0 > 2$. Thus, there is a non-empty set $A^*$ of solutions with $\frac{1}{2} \in A^*$ that yield positive $\alpha_0$ and positive $U_G$.

**Lemma 7.3.** *If $r'(\alpha_0) > \lambda_1 \phi_1'(\alpha_0)$ for a constant $\lambda_1 \geq 2$, then there exists a set $B^* \subseteq (0, 1)$ such that $\frac{1}{2} \in B^*$ and for all $\delta \in B^*$, $\alpha_1^*(\delta) > \alpha_0^*$, $U_D(\delta) > 0$.*

Let's presume $\delta^* = \frac{1}{2}$. If we can show that this solution yields $\alpha_1^* > \alpha_0^*$, then this means the domain specialist's utility is greater than 0 for investing some non-zero effort spend. $\alpha_1^*$ is the value of $\alpha_1$ that maximizes $U_D$, so if $U_D$ has positive slope at $\alpha_1 = \alpha_0^*$, then $\alpha_1^* > \alpha_0^*$.

Notice that this positive-utility outcome is met as long as $\frac{\partial U_D}{\partial \alpha_1}\big|_{\alpha_1=\alpha_0^*} > 0$ – this condition would necessarily mean that there exists some $\alpha_1^* > \alpha_0^*$ which maximizes $U_D$. So, formally:

$$r'(\alpha_0) > 2\phi_1'(\alpha_0)$$

$$\frac{1}{2}r'(\alpha_0) - \phi_1'(\alpha_0) > 0$$

$$\frac{\partial}{\partial \alpha_1}((1-\delta)r - \phi_1)\Big|_{\alpha_1=\alpha_0} > 0$$

$$\frac{\partial U_D}{\partial \alpha_1}\Big|_{\alpha_1=\alpha_0} > 0.$$

Notice that this inequality is met as long as $r'(\alpha_0) > \lambda_1 \phi_1'(\alpha_0)$ where $\lambda_1 \geq 2$. Thus, there is a non-empty set $B^*$ of solutions with $\frac{1}{2} \in B^*$ that yield $\alpha_1^* > \alpha_0^*$ and positive $U_D$.

**Corollary 7.1.** *$A^* \cap B^*$ is a non-empty set where any solution $\delta^* \in A^* \cap B^*$ satisfies the three properties: $0 < \delta_i < 1$; $0 < \alpha_0^* < \alpha_1^*$; and $U_D, U_G > 0$.*

This Corollary follows from the findings that have already been shown in the former Lemmas.

First, $A^* \cap B^*$ is non-empty because $\frac{1}{2} \in A^*$ (as shown in Lemma 7.2) and $\frac{1}{2} \in B^*$ (as shown in Lemma 7.3) so it follows that $\frac{1}{2} \in A^* \cap B^*$.

The three properties are each met for the following reasons:

(1) Property 1 ($0 < \delta^* < 1$) is met because $\frac{1}{2} \in A^* \cap B^*$ and $0 < \frac{1}{2} < 1$ so the existence finding is satisfied for a non-extreme value of $\delta$. This means, for $\delta^* \in A^* \cap B^*$, players share revenue.

(2) Property 2 ($0 < \alpha_0^* < \alpha_1^*$) is met because any solution in $A^*$ yields $\alpha_0^* > 0$ (as shown in Lemma 7.2) and any solution in $B^*$ yields $\alpha_1^* > \alpha_0^*$ (as shown in Lemma 7.3). This means, for $\delta^* \in A^* \cap B^*$, players do not free-ride (they both act to improve the technology).

(3) Property 3 ($U_G, U_D > 0$) is met because any solution in $A^*$ yields $U_G > 0$ (as shown in Lemma 7.2) and any solution in $B^*$ yields $U_D > 0$ (as shown in Lemma 7.3). This means any solution $\delta^* \in A^* \cap B^*$ yields positive utility for both players, which Pareto-dominates the disagreement scenario in which both players have zero utility.

□

The intuition for the above proof is that, as long as it is marginally profitable for both players to invest some finite positive amount of effort, then both players have positive utility (Pareto-dominating the disagreement alternative), neither player free rides, and both receive some profit.

This theorem perhaps helps explain why real-world situations arise where two different entities collaboratively invest in technologies. Cooperation is advantageous in a wide variety of fine-tuning scenarios. In the next section, we will further generalize our findings to a new set of games, for which the above theorems may be adapted.

## 8 SECTION 5 MATERIALS

### 8.1 Theorem on the Three Specialist Regimes

PROOF OF THEOREM 4.1. We prove this theorem in a sequence of Lemmas. The proof follows for any given specialist $D_i$ and revenue-sharing parameter $\delta$.

LEMMA 8.1. *If fixed costs are under control, meaning* $r_i(\alpha_0) > \frac{1}{1-\delta}\phi_i(\alpha_0)$, *then* $D_i$ *will not abstain – instead,* $D_i$ *would always prefer to free-ride.*

If $r_i(\alpha_0) > \frac{1}{1-\delta}\phi_i(\alpha_0)$, then $U_{D_i}\big|_{\alpha_i=\alpha_0} = r_i(\alpha_0) - \frac{1}{1-\delta}\phi_i(\alpha_0)$ is simply the RHS minus the LHS of the inequality. This means $U_{D_i}$ must be positive at $\alpha_i = \alpha_0$. Thus, as long as fixed costs are under control, the specialist prefers free-riding to abstaining.

LEMMA 8.2. *If fixed costs are not under control, meaning* $r_i(\alpha_0) < \frac{1}{1-\delta}\phi_i(\alpha_0)$, *then* $D_i$ *will not free-ride – instead,* $D_i$ *would always prefer to abstain.*

If $r_i(\alpha_0) < \frac{1}{1-\delta}\phi_i(\alpha_0)$, then $U_{D_i}\big|_{\alpha_i=\alpha_0} = r_i(\alpha_0) - \frac{1}{1-\delta}\phi_i(\alpha_0)$ is simply the RHS minus the LHS of the inequality. This means $U_{D_i}$ must be negative at $\alpha_i = \alpha_0$. Thus, as long as fixed costs are not under control, the specialist prefers abstaining to free-riding.

LEMMA 8.3. *If it is marginally profitable to invest in the technology, meaning* $r_i'(\alpha_0) > \frac{1}{1-\delta}\phi_i'(\alpha_0)$, *then* $D_i$ *will not free-ride – instead,* $D_i$ *would always prefer to contribute.*

If $r_i'(\alpha_0) > \frac{1}{1-\delta}\phi_i'(\alpha_0)$, then $\frac{\partial U_{D_i}}{\partial \alpha_i}\big|_{\alpha_i=\alpha_0} = r_i'(\alpha_0) - \frac{1}{1-\delta}\phi_i'(\alpha_0)$ is simply the RHS minus the LHS of the inequality. This means $U_{D_i}$ is increasing at $\alpha_i = \alpha_0$. Thus, as long as it is marginally profitable to improve the technology, the specialist prefers contributing to free-riding.

LEMMA 8.4. *If it is marginally costly to invest in the technology, meaning* $r_i'(\alpha_0) < \frac{1}{1-\delta}\phi_i'(\alpha_0)$, *then* $D_i$ *will not contribute – instead,* $D_i$ *would always prefer to free-ride.*

If $r_i'(\alpha_0) < \frac{1}{1-\delta}\phi_i'(\alpha_0)$, then $\frac{\partial U_{D_i}}{\partial \alpha_i}\big|_{\alpha_i=\alpha_0} = r_i'(\alpha_0) - \frac{1}{1-\delta}\phi_i'(\alpha_0)$ is simply the RHS minus the LHS of the inequality. This means $U_{D_i}$ is decreasing at $\alpha_i = \alpha_0$. Thus, as long as it is marginally costly to improve the technology, the specialist prefers free-riding to contributing.

Taken together, we can conclude the following about combinations of conditions:

- Fixed costs under control, marginally profitable investment: A<F, F<C (Lemmas 8.1 and 8.3). Thus the specialist would contribute.
- Fixed costs under control, marginally costly: A<F, C<F (Lemmas 8.1 and 8.4). Thus the specialist would free-ride.
- Fixed costs not under control, marginally profitable: F<A, F<C (Lemmas 8.2 and 8.3). Thus the specialist would either abstain or contribute.
- Fixed costs not under control, marginally costly: F<A, C<F (Lemmas 8.2 and 8.4). Thus the specialist would abstain.

Above, the short-hand notation 'A,' 'F,' and 'C' refer to the strategies of abstaining, free-riding, and contributing, respectively. The optimal strategies follow from the two marginal conditions. This completes the proof.

□

