# OpenReview forum: "Fine-Tuning Games: Bargaining and Adaptation for General-Purpose Models"
_ACM.org/TheWebConf/2024/Conference — TheWebConf24 Oral_

### Official Review · Reviewer_fs3H · 2023-11-24

**Novelty:** 6
**Technical Quality:** 5

**Review:**

The authors proposed a fine-tuning game model that captures the game between a general-purpose technology producer and one or more domain specialist(s). During bargaining stage, players bargain over how to share revenue, which is dependent on the extent of technology development; and then during the production stage, each role of the agent takes a turn to contribute to the technology’s performance. For unimodal utility function and polynomial cost function, the authors characterize different bargaining solutions under various objectives.

Overall, I enjoy reading the paper and the proposed two-stage model is intuitive. One of my question/concern is on the generality/sensitivity of the model and bargaining solutions to the assumptions of linear revenue function and polynomial cost functions, esp. such function with k_0, k_1>2 reflecting only marginally increasing cost in development effort. It would be helpful if the authors could further characterize what kind of cost functions can lead to unimodal utility. Another suggestion is to add more interpretations to the results and the bargaining solution, e.g., why would the case of maximizing tech performance lead to a single role of agent (either general producer or specialist) involving in the development process? See below for more detailed questions.

**Questions:**

1. Could the authors please provide some intuition on why would the case of maximizing tech performance lead to a single role of agent (either general producer or specialist) involving in the development process? Is this a result of the polynomial cost function?

2. Could the authors provide some other class of cost functions that gives an unimodal utility function?

3. I didn't fully understand the categorization of regimes of strategies, i.e., contributor, free-rider, and abstainer. For the specialist, is being a contributor/free-rider always a weakly dominant strategy to being an abstainer? I assume that the specialist receives utility 0 whenever there's a disagreement. Could the specialist still develop its own product after abstain?

**Reviewer Confidence:**

3: The reviewer is confident but not certain that the evaluation is correct

**Scope:**

4: The work is relevant to the Web and to the track, and is of broad interest to the community

---

### Official Review · Reviewer_mQdo · 2023-11-28

**Novelty:** 4
**Technical Quality:** 2

**Review:**

Summary

This paper considers a class of bargaining games inspired by the relationship that happens when one party fine-tunes an AI model produced by a second party. Specifically, consider a setting where one party (the "generalist") originally trains a model to quality x_1 at a cost of c_G(x_1); the second party (the "domain specialist") then fine-tunes this model to quality x_2, at a cost of c_D(x_1, x_2). The generalist and domain specialist split the revenue of this final model (assumed to just be equal to x_2) according to a fixed revenue split delta, decided on at the beginning of the game.

Each delta fixes an outcome of this game (a specific action and eventual utility for both parties). The authors examine the Pareto frontier of this set of outcomes and point out several specific bargaining solutions on this frontier (e.g. the outcome when one party gets to unilaterally set delta, the Nash bargaining solution, the max social welfare solution, ...). The authors then consider a multi-agent generalization of this problem where there are many different domain experts but a single fixed delta. The authors provide simple conditions to decide whether a domain expert in this setting will be a "free rider" (not improve the model at all), a "contributor" (improve the model), or an "abstainer" (not participate in this deal with the generalist).

Evaluation

Games where two parties iterate on a solution (such as fine-tuning an AI model) are a natural class of games and increasingly important with the increasing prevalence of LLMs. This paper proposes an interesting concrete economic model for understanding such games, and presents some understanding of the space of solutions for these games.

While I find the model reasonably interesting, I think the paper lacks a bit in substantive results / insights to draw from it. It is somewhat clear a priori that (in the two-party case) there is going to be a range of Pareto-optimal delta, and that different delta correspond to different tradeoffs on the Pareto-frontier. The multi-agent case is a little more interesting, but it is still not too surprising that agents will fall into one of these categories under certain assumptions. Perhaps it would be more interesting to understand how the generalist should choose delta in a multi-agent setting, or to understand general tradeoffs in a quantitative way (e.g. how big is the gap in social welfare between the best and worst point on the P-O curve? are there any natural conditions where it is small? when can the generalist benefit by charging a fixed cost for access to the model instead of a revenue share?)

**Questions:**

Feel free to respond to any comments / potential misunderstandings in the review above.

**Reviewer Confidence:**

3: The reviewer is confident but not certain that the evaluation is correct

**Scope:**

3: The work is somewhat relevant to the Web and to the track, and is of narrow interest to a sub-community

---

### Official Review · Reviewer_hNew · 2023-12-02

**Novelty:** 6
**Technical Quality:** 5

**Review:**

This paper investigates the game behind the business of fine-tuning general-purpose AI and ML models. It introduces a model illustrating the dynamics between generalists (developers of the technology) and domain specialists (adapters for specific uses), focusing on the bargaining and revenue-sharing aspects among stakeholders in technology adaptation.

## Quality

This paper points out a significant and increasingly relevant area in the application of AI. It provides an interesting understanding of the economic and strategic considerations in AI technology adaptation and deployment.

My primary concern is the rationale behind modeling the fine-tuning game as presented in the paper. In the current landscape, generalists have already developed comprehensive models (e.g. GPT4 and Lllma2). The challenge, however, lies in bargaining to get domain specialists to further adapt and refine these models for specific domain applications. (e.g. GPTs https://openai.com/blog/introducing-gpts)

In addition, the paper assumes that in a disagreement scenario, both the specialist and the generalist have 0 revenue. However, it also admits that “it could be the case that a general-purpose technology producer receives positive payout even if the specialist abstains from a bargain”. I’m wondering why not assume that the generalist has $r(\alpha_0)$ revenue in a disagreement scenario where $\alpha_0$ is the performance of the base model.

Another minor question is why the agents are not necessarily individually rational, i.e. why they still play the game even if their utility is negative.


## Clarity

Overall, the paper is well-written and clear, offering detailed explanations and articulations of its concepts, making it accessible to readers with varying levels of expertise in the field.

## Originality

The approach to model the interaction between generalists and specialists in the context of AI and ML adaptation fills a gap in the existing literature by providing a structured game theory framework to understand the fine-tuning of general-purpose models in specific domains.

My concerns may be raised since the paper lacks an extensive literature review on bargaining games, particularly those involving joint production, which seems to be highly relevant to the topic. Including such a review might strengthen the theoretical foundation of the paper and provide a more transparent view of the paper’s originality.

## Significance

The research question is significant as it explores the strategic behavior of firms in technology adaptation, contributing to both academic research and practical applications in AI and ML.

## Other comments

An interesting area for future exploration is to generalize the current bargaining game to a multi-round game to reflect the rapid and iterative development of AI and ML technologies.

In addition, exploring competition scenarios involving multiple generalists or multiple specialists in one domain might also be interesting. This might be a feasible approach to understanding that “players do not necessarily opt to maximize their own share of the profit (Line 674)”.

## Minor Issues

A minor issue is that the use of the term “multi-specialist fine-tuning game” (line 746) seems to be a little misleading for me, as the paper deals with multi-domain cases without competition among specialists.

Line 340, “strictly unimodal function over a real domain D”

**Questions:**

See the review above

**Reviewer Confidence:**

3: The reviewer is confident but not certain that the evaluation is correct

**Scope:**

4: The work is relevant to the Web and to the track, and is of broad interest to the community

---

### Decision · Program_Chairs · 2024-01-22

**Decision:**

Accept (Oral)

**Comment:**

The submitted paper studies a game-theoretic setting that illustrates the dynamics underlying the development/fine-tuning of general-purpose AI/ML models. Specifically, it considers a setting with ``generalists'' who develop a general-purpose model and domain specialists who fine-tune it to a particular application.

 The paper provides an interesting model to study a setting that is of great economic importance at this point in time. That, I think, is its great contribution (also reflected in two novelty scores of 6). At the same time, the review team was not fully convinced of the technical contribution relative to the existing bargaining literature (i.e., is this setting, on a technical level, really so different from traditional bargaining/supply chain settings?).

 Accordingly, the reviewers were not fully aligned on the contribution of the paper: one gave a low technical score (and borderline novelty score), two gave high novelty scores and reasonable technical scores. I am inclined to side with the higher scores, in particular as the authors' rebuttal included a detailed response to the first reviewer, which may address some of that reviewer's concerns.